# NeoBERT: A Next-Generation BERT

**Lola Le Breton**[1,2,3]    **Quentin Fournier**[2]    **John X. Morris**[4]    **Mariam El Mezouar**[5]
**Sarath Chandar**[1,2,3,6]

[1]Chandar Research Lab    [2]Mila – Quebec AI Institute    [3]Polytechnique Montréal
[4]Cornell University    [5]Royal Military College of Canada    [6]Canada CIFAR AI Chair

**Reviewed on OpenReview:** `https://openreview.net/forum?id=TJRyDi7mwH`

## Abstract

Recent innovations in architecture, pre-training, and fine-tuning have led to the remarkable in-context learning and reasoning abilities of large auto-regressive language models such as LLaMA and DeepSeek. In contrast, encoders like BERT and RoBERTa have not seen the same level of progress despite being foundational for many downstream NLP applications. To bridge this gap, we introduce NeoBERT, a next-generation encoder that redefines the capabilities of bidirectional models by integrating state-of-the-art advancements in architecture, modern data, and optimized pre-training methodologies. NeoBERT is designed for seamless adoption: it serves as a plug-and-play replacement for existing base models, relies on an optimal depth-to-width ratio, and leverages an extended context length of 4,096 tokens. Despite its compact 250M parameter footprint, it achieves state-of-the-art results on the massive MTEB benchmark, outperforming $BERT_{large}$, $RoBERTa_{large}$, NomicBERT, and ModernBERT under identical fine-tuning conditions. In addition, we rigorously evaluate the impact of each modification on GLUE and design a uniform fine-tuning and evaluation framework for MTEB. We release all code, data, checkpoints, and training scripts to accelerate research and real-world adoption [1,2].

## 1 Introduction

Auto-regressive language models have made tremendous progress since the introduction of GPT (Radford et al., 2018), and modern large language models (LLMs) such as LLaMA 3 (Dubey et al., 2024), Mistral (Jiang et al., 2023), OLMo (Groeneveld et al., 2024), and DeepSeek-r1 (DeepSeek-AI et al., 2025) now exhibit remarkable reasoning and in-context learning capabilities. These improvements result from scaling both the models and the web-scraped text datasets they are trained on, as well as from innovations in architecture and optimization techniques. However, while decoders have continuously evolved, encoders have not seen the same level of progress. Worse, their knowledge has become increasingly outdated despite remaining critical for a wide range of downstream NLP tasks that depend on their embeddings, notably for retrieval-augmented generation (Ram et al., 2023) and toxicity classification (Hartvigsen et al., 2022). Despite being five years old, BERT (Devlin et al., 2019) and RoBERTa (Liu et al., 2019) remain widely used, with more than 110 million combined downloads from Hugging Face as of the writing of this paper.

Similar to decoders, which undergo multi-stage processes of pre-training, instruction-tuning, and alignment, encoders also require successive training phases to perform well on downstream tasks. First, encoders go through self-supervised pre-training on large corpora of text with the masked language modeling objective. By predicting masked or replaced tokens, this stage enables models to learn the structural patterns of language and the semantics of words. However, the pre-training task is disconnected from downstream applications, and models require additional training to achieve strong performance in clustering or retrieval.

---

[1] `https://huggingface.co/chandar-lab/NeoBERT`
[2] `https://github.com/chandar-lab/NeoBERT`

Thus, a second fine-tuning phase is often achieved through multiple stages of semi-supervised contrastive learning, where models learn to differentiate between positive and negative sentence pairs, refining their embeddings in the latent space.

Recently, substantial progress has been made in improving the fine-tuning stage of pre-trained encoders, with models like GTE (Li et al., 2023b), jina-embeddings (Sturua et al., 2024), SFR-embeddings (Meng et al., 2024), and CDE (Morris & Rush, 2024) significantly outperforming previous encoders on the MTEB leaderboard, a recent and challenging benchmark spanning 7 tasks and 56 datasets. However, all these approaches focus on proposing complex fine-tuning methods and do not address the inherent limitations of their pre-trained backbone models.

As a result, there is a dire need for a new generation of BERT-like pre-trained models that incorporate up-to-date knowledge and leverage both architectural and training innovations, forming stronger backbones for these more advanced fine-tuning procedures.

In response, we introduce NeoBERT, a next-generation BERT that integrates the latest advancements in architecture, data, and pre-training strategies. The improvements are rigorously validated on GLUE by fully pre-training 10 different models that successively incorporate the modifications. This validation ensures that the improvements benefit encoder architectures and highlights how some design choices drastically affect the model's abilities. Additionally, we design and experimentally validate a two-stage training procedure to increase NeoBERT's maximum context window from $1,024$ to $4,096$. To ensure a fair evaluation of NeoBERT against existing baselines and to isolate the impact of fine-tuning procedures, we propose a model-agnostic and systematic fine-tuning strategy with straightforward contrastive learning. All models are fine-tuned using this standardized approach and subsequently evaluated on the MTEB benchmark.

On MTEB, NeoBERT consistently outperforms all competing pre-trained models while being 100M parameters smaller than the typical *large*-sized encoders. With a context window of 4,096 tokens, it processes sequences $8\times$ longer than RoBERTa (Liu et al., 2019) and two times longer than NomicBERT (Nussbaum et al., 2024). It is also the fastest encoder of its kind, significantly outperforming ModernBERT *base* and *large* in terms of inference speed. Despite its compact 250M parameter size, NeoBERT is trained for over 2T tokens, prioritizing training over scale to maximize accessibility for both academic researchers and industry users without requiring large-scale compute resources. This makes NeoBERT the most extensively trained model among modern encoders, ensuring robust generalization and superior downstream performance. Furthermore, NeoBERT maintains the same hidden size as *base* models, allowing for seamless plug-and-play adoption without modifications to existing architectures. As the only fully open-source model of its kind, we release the code, data, training scripts, and model checkpoints, reinforcing our commitment to reproducible research.

## 2 Related work

In 2019, Devlin et al. (2019) introduced BERT, a novel approach to embedding text using bi-directional Transformers pre-trained without supervision on large corpora. Shortly after, Liu et al. (2019) improved over BERT's pre-training by removing the next-sentence prediction objective and drastically increasing the amount of data, leading to RoBERTa. Since then, the primary focus of the community has shifted towards optimizing the fine-tuning phase of these models through contrastive learning, where the model is trained to maximize the similarity between positive text pairs while pushing them apart from negative samples.

Among the earliest contrastive learning approaches designed for encoders, SimCSE (Gao et al., 2022) demonstrated that sentence pairs could be easily generated by feeding the same input to the model twice and applying dropout to introduce noise. However, this simple approach was soon outperformed by models like GTE (Li et al., 2023b), which introduced more advanced contrastive learning techniques. GTE employed a weakly supervised stage that takes advantage of the vast number of successive sentence pairs available in traditional unlabeled datasets, followed by a semi-supervised stage incorporating labeled sentence pairs from high-quality datasets such as NLI (Bowman et al., 2015) and FEVER (Thorne et al., 2018). Recently, fine-grained strategies have emerged to better adapt models to both task and context. For instance, Jina-embeddings (Sturua et al., 2024) introduced task-specific Low-Rank Adaptation (LoRA) adapters. As of

January 2025, CDE (Morris & Rush, 2024) ranks at the top of the MTEB leaderboard for models under 250M parameters thanks to two key innovations: grouping samples with related contexts into the same batch and providing contextual embeddings for the entire corpus in response to individual queries.

However, pre-training has not seen the same level of effort, and thus progress, most likely due to its prohibitively high computational cost. RoBERTa, for instance, required a total of $1,024$ V100 days for its pre-training. As a result, GTE, Jina-embeddings, and CDE all rely on pre-trained BERT, XLM-RoBERTa (Conneau et al., 2020), and NomicBERT (Nussbaum et al., 2024) to initialize their respective models. The latter, NomicBERT, represents a recent effort to refine BERT's architecture and pre-training. NomicBERT incorporates architectural improvements such as SwiGLU and RoPE, utilizes FlashAttention, and extends the context length to $2,048$ tokens. Despite these innovations, NomicBERT still relied on sub-optimal choices, as discussed in section 3. In parallel with the development of NeoBERT, Warner et al. (2024) released ModernBERT with the goal of further refining the pre-training of NomicBERT. Although we share some of the modifications, we make distinct design choices and conduct thorough ablations that ultimately lead to greater performance on MTEB.

## 3  NeoBERT

The following section describes NeoBERT's improvements over BERT and RoBERTa, as well as the recent NomicBERT and ModernBERT models. Since GTE and CDE use BERT and NomicBERT as their pre-trained backbone, they inherit their respective characteristics. Table 1 summarizes the modifications.

Table 1: Comparison of Model Architectures, Training Data, and Pre-Training Configurations.

| | BERT | | RoBERTa | | NomicBERT | ModernBERT | | NeoBERT |
|---|---|---|---|---|---|---|---|---|
| | *base* | *large* | *base* | *large* | *base* | *base* | *large* | *medium* |
| **Layers** | 12 | 24 | 12 | 24 | 12 | 22 | 28 | 28 |
| **Hidden Size** | 768 | 1,024 | 768 | 1,024 | 768 | 768 | 1,024 | 768 |
| **Attention Heads** | 12 | 16 | 12 | 16 | 12 | 12 | 16 | 12 |
| **Parameters** | 120M | 350M | 125M | 355M | 137M | 149M | 395M | 250M |
| **Activation Function** | GeLU | | | | SwiGLU | GeGLU | | SwiGLU |
| **Positional Encoding** | Positional Embeddings | | | | RoPE | RoPE | | RoPE |
| **Normalization** | Post-LayerNorm | | | | Post-LayerNorm | Pre-LayerNorm | | Pre-RMSNorm |
| **Data Sources** | BooksCorpus Wikipedia | | BooksCorpus OpenWebText Stories / CC-News | | BooksCorpus Wikipedia | Undisclosed | | RefinedWeb |
| **Dataset Size** | 13GB | | 160GB | | 13GB | - | | 2.8TB |
| **Dataset Year** | 2019 | | 2019 | | 2023 | - | | 2023 |
| **Tokenizer Level** | Character | | Byte | | Character | Character | | Character |
| **Vocabulary Size** | 30K | | 50K | | 30K | 50K | | 30K |
| **Sequence Length** | 512 | | 512 | | 2,048 | $1,024 \rightarrow 8,192$ | | $1,024 \rightarrow 4,096$ |
| **Objective** | MLM + NSP | | MLM | | MLM | MLM | | MLM |
| **Masking Rate** | 15% | | 15% | | 30% | 30% | | 20% |
| **Masking Scheme** | 80/10/10 | | 80/10/10 | | - | - | | 100 |
| **Optimizer** | Adam | | Adam | | AdamW | StableAdamW | | AdamW |
| **Scheduler** | - | | - | | - | WSD | | CosineDecay |
| **Batch Size** | 131k tokens | | 131k | | 8M | 448k to 5M | | 2M |
| **Tokens Seen** | 131B | | 131B | | - | $\sim 2T$ | | 2.1T |
| **Training** | DDP | | DDP | | DeepSpeed FlashAttention | Alternate Attention Unpadding FlashAttention | | DeepSpeed FlashAttention |

### 3.1 Architecture

The Transformer architecture has been refined over the years and has now largely stabilized, with models like LLaMA 3 being essentially the same as the original LLaMA. NeoBERT integrates the latest modifications that have, for the most part, become standard.

**Depth-to-Width**  The concept of depth efficiency has long been recognized in neural network architectures. In the case of Transformers, stacking self-attention layers is so effective that it can quickly saturate the network's capacity. Recognizing this, Levine et al. (2020) provided theoretical and empirical evidence for an optimal depth-to-width ratio in Transformers. Their findings suggested that most language models were operating in a "depth-inefficiency" regime, where allocating more parameters to width rather than depth would have improved performance. In contrast, small language models like BERT, RoBERTa, and NomicBERT are instead in a width-inefficiency regime. To maximize NeoBERT's parameter efficiency while ensuring it remains a seamless plug-and-play replacement, we retain the original $\text{BERT}_{base}$ width of 768 and instead increase its depth to achieve this optimal ratio.

**Positional Information**  Transformers inherently lack the ability to distinguish token positions. Early models like BERT and RoBERTa addressed this by adding absolute positional embeddings to the token embeddings before the first Transformer block. However, this approach struggles to generalize to longer sequences and requires the positional information to be propagated across layers. To overcome these limitations, Su et al. (2023) proposed Rotary Position Embeddings (RoPE), which integrate relative positional information directly into the self-attention mechanism. RoPE has quickly become the default in modern Transformers due to its significant improvements in performance and extrapolation capabilities. NeoBERT, like all newer encoders, integrates RoPE. Nevertheless, degradation still occurs with sequences significantly longer than those seen during training. As a solution, Peng et al. (2023) introduced Yet Another RoPE Extension (YaRN), which allows to efficiently fine-tune models on longer contexts. NeoBERT is readily compatible with YaRN, making it well-suited for tasks requiring extended context.

**Layer Normalization**  Consistent with standard practices in modern Transformer architectures, we move the normalization layer inside the residual connections of each feed-forward and attention block, a technique known as Pre-Layer Normalization (Pre-LN). Pre-LN improves stability, allows for larger learning rates, and accelerates model convergence (Xiong et al., 2020). While all newer encoder models adopt Pre-LN, they typically continue to use the classical LayerNorm rather than Root Mean Square Layer Normalization (RMSNorm). In NeoBERT, we substitute the classical LayerNorm with RMSNorm (Zhang & Sennrich, 2019), which achieves comparable training stability while being slightly less computationally intensive, as it requires one fewer statistic.

**Activations**  BERT and RoBERTa utilize the standard Gaussian Error Linear Unit (GELU) activation function. However, Shazeer (2020) demonstrated the benefits of the Gated Linear Unit in Transformer architectures. These activation functions have since been adopted in several language models, including the LLaMA family. Following previous works, NeoBERT incorporates the SwiGLU activation function, and because it introduces a third weight matrix, we scale the number of hidden units by a factor of $\frac{2}{3}$ to keep the number of parameters constant.

### 3.2 Data

Data has emerged as one of the most critical aspects of pre-training, and datasets with increasing quantity and diversity are frequently released. NeoBERT takes advantage of the latest datasets that have proven to be effective.

**Dataset**  BERT and NomicBERT were pre-trained on two carefully curated and high-quality datasets: Wikipedia and BookCorpus (Zhu et al., 2015). As Baevski et al. (2019) demonstrated that increasing data size can improve downstream performance, Liu et al. (2019) pre-trained RoBERTa on 10 times more data from BookCorpus, CC-News, OpenWebText, and Stories. However, RoBERTa's pre-training corpus has

become small in comparison to modern web-scraped datasets built by filtering and deduplicating Common Crawl dumps. Following the same trend, we pre-trained NeoBERT on RefinedWeb (Penedo et al., 2023), a massive dataset containing 600B tokens, nearly 18 times larger than RoBERTa's. Although RefinedWeb does not have strict high-quality constraints, we believe that exposing the model to such a large and diverse dataset will improve its real-world utility.

**Sequence Length**  BERT and RoBERTa were pre-trained on sequences up to 512 tokens, which limits their downstream utility, especially without RoPE and YaRN. NomicBERT increased the maximum length to $2,048$ and employed Dynamic NTK interpolation at inference to scale to 8192. To further broaden NeoBERT's utility, we seek to increase the context length. However, due to the computational cost associated with pre-training, we adopt a two-stage pre-training procedure similar to LLMs like LLaMA 3. In the first stage, we train the model for 1M steps (2T tokens) using sequences truncated to a maximum length of $1,024$ tokens, referring to this version as $\text{NeoBERT}_{1024}$. In the second stage, we extend the training for an additional 50k steps (100B tokens), increasing the maximum sequence length to $4,096$ tokens. We refer to this final model as $\text{NeoBERT}_{4096}$. To ensure the model encounters longer sequences during this stage, we create two additional sub-datasets, $\text{Refinedweb}_{1024+}$ and $\text{Refinedweb}_{2048+}$, containing only sequence lengths greater than $1,024$ and $2,048$ tokens, respectively, alongside the original Refinedweb dataset. Each batch is sampled from Refinedweb, $\text{Refinedweb}_{1024+}$ and $\text{Refinedweb}_{2048+}$ with probabilities 20%, 40%, and 40%. Since longer sequences tend to represent more complex or academic content, this strategy helps mitigate the distribution shift typically observed when filtering for longer sequences. We explore the benefits of this two-stage training strategy in subsection 5.3.

### 3.3  Pre-Training

Encoder pre-training has received less attention than the data and architecture. However, many improvements made to decoders also apply to encoders. NeoBERT combines encoder-specific modifications with widely accepted decoder improvements.

**Objective**  In light of RoBERTa's findings that dropping the next-sentence prediction task does not harm performance, both NomicBERT and NeoBERT were only pre-trained on masked language modeling. Moreover, Wettig et al. (2023) challenged the assumption that the 15% masking rate of BERT and RoBERTa is universally optimal. Instead, their findings suggest that the optimal masking rate is actually 20% for base models and 40% for large models. Intuitively, a model learns best when the difficulty of its training tasks aligns with its capabilities. Based on their insight, we increase the masking rate to 20%, while NomicBERT exceeds it by opting for 30%.

**Optimization**  Following standard practice, we use the AdamW optimizer (Loshchilov & Hutter, 2019) with the same hyperparameters as LLaMA 2: $\beta_1 = 0.9$, $\beta_2 = 0.95$, and $\epsilon = 10^{-8}$. In preliminary experiments, we also considered SOAP (Vyas et al., 2025), a recent extension of the Shampoo optimizer, but it failed to outperform Adam and AdamW and has been omitted from the list of ablations. We employ a linear warmup for $2,000$ steps to reach a peak learning rate of $6 \times 10^{-4}$, followed by a cosine decay to 10% of the peak learning rate over 90% of the training steps. Once fully decayed, the learning rate remains constant for the last 100k steps at a sequence length of $1,024$ and 50k steps at a sequence length of $4,096$. We use a weight decay of 0.1 and apply gradient clipping with a maximum norm of 1.0.

**Scale**  Recent generative models like the LLaMA family (Touvron et al., 2023; Dubey et al., 2024) have demonstrated that language models benefit from being trained on significantly more tokens than was previously standard. Recently, LLaMA-3.2 1B was successfully trained on up to 9T tokens without showing signs of saturation. Moreover, encoders are less sample-efficient than decoders since they only make predictions for masked tokens. Therefore, it is reasonable to believe that encoders of similar sizes can be trained on an equal or even greater number of tokens without saturating. For NeoBERT's pre-training, we use batch sizes of 2M tokens over 1M steps in the first stage and 50k steps in the second, resulting in a theoretical total of 2.1T tokens. Note that because sequences are padded to the maximum length, this represents a theoretical number of tokens. In terms of tokens, this is comparable to RoBERTa and represents a 2x increase over

Table 2: Modifications between successive ablations. The initial $M0$ baseline corresponds to a model similar to BERT, while $M9$ corresponds to NeoBERT.

|  | Modification | Before | After |
|---|---|---|---|
| $M1$ | Embedding
Activation
Pre-LN | Positional
GELU
LayerNorm | RoPE
SwiGLU
RMSNorm |
| $M2$ | Dataset | Wiki + Book | RefinedWeb |
| $M3$ | Tokenizer | Google WordPiece | LLaMA BPE |
| $M4$ | Optimizer
Scheduler | Adam
Linear | AdamW
Cosine |
| $M5$ | Masking Scheme | 15% (80 / 10 / 10) | 20% (100) |
| $M6$ | Sequence packing | False | True |
| $M7$ | Model Size | 120M | 250M |
| $M8$ | Depth - Width | 16 - 1056 | 28 - 768 |
| $M9$ | Batch size
Context length | 131k
512 | 2M
4, 096 |

NomicBERT. In terms of training steps, this amounts to a 2x increase over RoBERTa and a 10x increase over NomicBERT.

**Efficiency**   We improve efficiency by parallelizing the model across devices using DeepSpeed (Aminabadi et al., 2022) with the ZeRO (Rajbhandari et al., 2020) optimizer, reducing memory usage by eliminating data duplication across GPUs and increasing the batch size. We further optimize the GPU utilization by employing fused operators from the `xFormers` library to reduce overhead, selecting all dimensions to be multiples of 64 to align with GPU architectures, and removing biases to simplify computations without sacrificing performance. To address the quadratic demands of attention, we integrate FlashAttention (Dao, 2023), which computes exact attention without storing the full matrices.

## 4   Effect of Design Choices

We conduct a series of ablations in controlled settings to evaluate our improvements to the original BERT architecture. We fully train each model for 1M steps, controlling for the seed and dataloader states to ensure successive models are trained with identical setups. These resource-intensive ablations were crucial to confirm our design choices, as they are based on the literature of pre-training decoder models. The baseline model, referred to as $M0$, is similar to BERT$_{base}$ but includes pre-layer normalization. Following RoBERTa, $M0$ also drops the next sentence prediction objective. We introduce modifications iteratively, resulting in a total of ten different models, as detailed in Table 2. To mitigate computational costs, the ablations are evaluated on the GLUE benchmark with a limited hyperparameter grid search of batch sizes $\in \{16, 32\}$ and learning rates $\in \{1e-5, 2e-5, 3e-5\}$. For the final model $M10$, we extend the grid search, as detailed in Appendix C. Results are in Figure 1.

**Key Performance-Enhancing Modifications**   As expected, the two modifications that had the greatest impact on the average GLUE score were related to scale. In $M2$, replacing Wikitext and BookCorpus with the significantly larger and more diverse RefinedWeb dataset improved the score by +3.6%, while increasing the model size from 120M to 250M in $M7$ led to a +2.9% relative improvement. Note that to assess the impact of the depth-to-width ratio, we first scale the number of parameters in $M7$ to 250M while maintaining

Figure 1: GLUE ablation scores on the development set. All modifications are cumulative, except for $M3$ and $M6$ in grey, which are not included in the subsequent models. Increasing data size and diversity leads to the highest relative improvement ($M2$, +3.6%), followed by the model size ($M7$, +2.9%). Packing the sequences and using the LLaMA 2 tokenizer cause the largest relative drops ($M6$, −2.9%, $M3$, −2.1%).

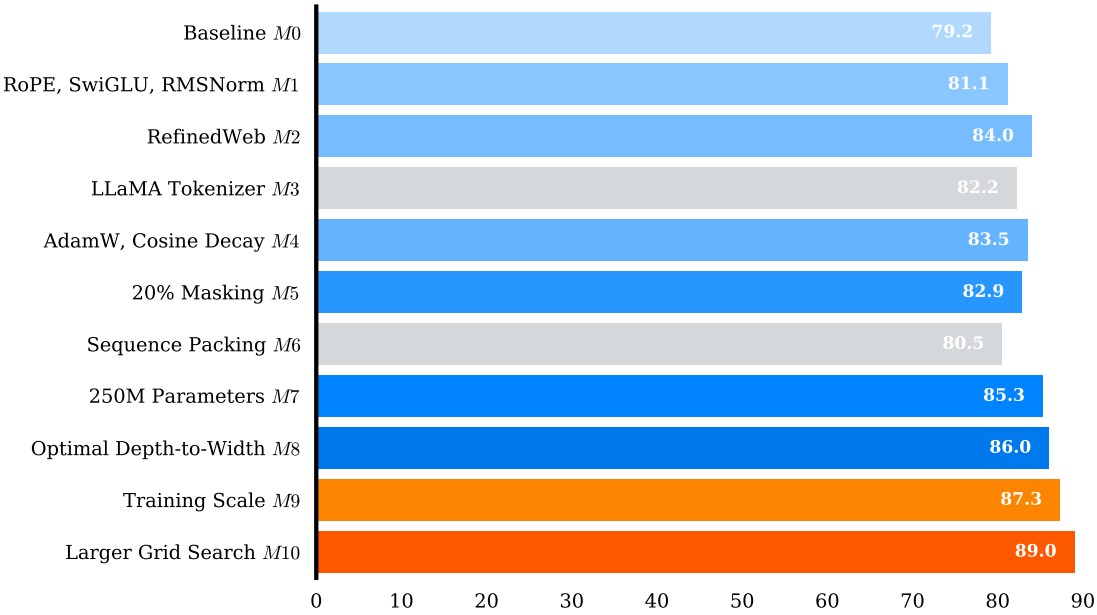

a similar ratio to $\text{BERT}_{base}$, resulting in 16 layers of dimension 1056. In $M8$, the ratio is then adjusted to 28 layers of dimension 768.

**Modifications That Were Discarded** In $M3$, replacing the Google WordPiece tokenizer with LLaMA BPE results in a −2.1% performance decrease. The performance gap observed likely stems from inherent differences between the algorithms, training corpus, and design strategies employed by the two tokenizers. We further discuss this in Appendix B. In $M6$, we unpad the sequences by concatenating samples of the same batch. While this removes unnecessary computation on padding tokens, packing sequences without accounting for cross-sequence attention results in a relative performance drop of −2.8%. Although this modification was discarded from our subsequent ablations, we incorporate methods of un-padding with accurate cross-attention in our released codebase, following Kundu et al. (2024).

**Modifications Retained Despite Performance Trade-offs** Unexpectedly, using AdamW (Loshchilov & Hutter, 2019) and cosine decay (Loshchilov & Hutter, 2017) in $M4$ decreases performance by −0.5%. As AdamW introduces additional regularization with weight decay, we expect that it will become beneficial when extending training by mitigating overfitting. Similarly, increasing the masking ratio from 15% to 20% in $M5$ decreases performance by −0.7%. We hypothesize that increasing the task difficulty initially hinders downstream task performance but is likely to become advantageous when training larger models on more tokens. Consequently, we retain both modifications despite being unable to verify these hypotheses at scale due to the computational costs.

## 5 Experiments

Selecting appropriate metrics and benchmarks is crucial for properly assessing the downstream performance and utility of language models. Following both early and recent studies, we include the GLUE and MTEB benchmarks in our evaluations.

## 5.1 GLUE

The GLUE benchmark (Wang et al., 2019) is a cornerstone of language modeling evaluations and has played a significant role in the field. Although GLUE is now 6 years old and the community has long recognized its limitations, we report the GLUE score due to its widespread adoption and to facilitate the comparison of NeoBERT with existing encoders. Following standard practices, we fine-tune NeoBERT on the development set of GLUE with a classical hyperparameter search and introduce transfer learning between related tasks. Complete details of the fine-tuning and best hyperparameters are presented in Appendix C. NeoBERT achieves a score of $89.0\%$ comparable to previous *large* models while being $100M$ to $150M$ parameters smaller. We present the results in Table 3.

Table 3: GLUE scores on the development set. Baseline scores were retrieved as follows: BERT from Table 1 of Devlin et al. (2019), RoBERTa from Table 8 of Liu et al. (2019), DeBERTa from Table 3 of He et al. (2023), NomicBERT from Table 2 of Nussbaum et al. (2024), GTE from Table 13 of Zhang et al. (2024), and ModernBERT from Table 5 of Warner et al. (2024).

| Size | Model | MNLI | QNLI | QQP | RTE | SST | MRPC | CoLA | STS | Avg. |
|------|-------|------|------|-----|-----|-----|------|------|-----|------|
| Base ($\leq 150M$) | BERT | 84.0 | 90.5 | 71.2 | 66.4 | 93.5 | 88.9 | 52.1 | 85.8 | 79.6 |
| | RoBERTa | 87.6 | 92.8 | 91.9 | 78.7 | 94.8 | 90.2 | 63.6 | 91.2 | 86.4 |
| | GTE-en-8192 | 86.7 | 91.9 | 88.8 | 84.8 | 93.3 | 92.1 | 57.0 | 90.2 | 85.6 |
| | NomicBERT$_{2048}$ | 86.0 | 92.0 | 92.0 | 82.0 | 93.0 | 88.0 | 50.0 | 90.0 | 84.0 |
| | ModernBERT | 89.1 | 93.9 | 92.1 | 87.4 | 96.0 | 92.2 | 65.1 | 91.8 | 88.5 |
| Medium $250M$ | NeoBERT$_{1024}$ | 88.9 | 93.9 | 90.7 | 91.0 | 95.8 | 93.4 | 64.8 | 92.1 | 88.8 |
| | NeoBERT$_{4096}$ | 89.0 | 93.7 | 90.7 | 91.3 | 95.6 | 93.4 | 66.2 | 91.8 | 89.0 |
| Large ($\geq 340M$) | BERT | 86.3 | 92.7 | 72.1 | 70.1 | 94.9 | 89.3 | 60.5 | 86.5 | 82.1 |
| | RoBERTa | 90.2 | 94.7 | 92.2 | 86.6 | 96.4 | 90.9 | 68.0 | 92.4 | 88.9 |
| | DeBERTaV3 | **91.9** | **96.0** | **93.0** | **92.7** | 96.9 | 91.9 | **75.3** | **93.0** | **91.4** |
| | GTE-en-8192 | 89.2 | 93.9 | 89.2 | 88.1 | 95.1 | **93.5** | 60.4 | 91.4 | 87.6 |
| | ModernBERT | 90.8 | 95.2 | 92.7 | 92.1 | **97.1** | 91.7 | 71.4 | 92.8 | 90.5 |

## 5.2 MTEB

Beyond the GLUE benchmark, we consider the more recent and challenging MTEB benchmark (Muennighoff et al., 2023), which has quickly become a standard for evaluating embedding models, with a wide coverage of 7 tasks and 56 datasets in its English subset.

MTEB tasks rely on the cosine similarity of embeddings pooled across tokens in a sentence. The most common and straightforward pooling strategy is computing the average of each token's final hidden representation. However, because out-of-the-box encoders are trained with the masked language modeling objective, they provide no guarantee that mean embeddings will produce meaningful representations without further fine-tuning. As a result, models require expensive fine-tuning strategies crafted for MTEB to achieve strong scores. For instance, GTE (Li et al., 2023b) with multi-stage contrastive learning, Jina-embeddings (Sturua et al., 2024) with task-specific Low-Rank Adaptation (LoRA) adapters, and CDE (Morris & Rush, 2024), with batch clustering and contextual corpus embeddings all pushed the limits of the leaderboard in their respective categories.

These coupled stages make it challenging to disentangle the respective impacts of pre-training and fine-tuning on the final model's performance. To isolate and fairly evaluate the improvements introduced during pre-training, we implemented an affordable, model-agnostic fine-tuning strategy based on classical contrastive learning. This fine-tuning approach was designed in accordance with established methods in the literature. Its controlled settings ensured that all models were fine-tuned and evaluated under identical conditions.

### 5.2.1 Unified Contrastive Learning

**Method**  Given a dataset of positive pairs $\mathbb{D} = \{q_i, d_i^+\}_{i=1}^n$, a similarity metric $s$, a temperature parameter $\tau$, and a set of negative documents $N_q$ for each query $q$, the contrastive loss is defined as:

$$\mathcal{L} = -\log \frac{e^{s(q,d^+)/\tau}}{e^{s(q,d^+)/\tau} + \sum_{d^- \in N_q} e^{s(q,d^-)/\tau}}$$

Negative documents can be either generic samples of the same format or tailored hard negatives, which exhibit a high degree of similarity to the contrasted sample in their original representation. We constructed a dataset of positive query-document pairs with optional hard negatives based on open-source datasets previously employed in the literature (Li et al., 2023a) for a total of nine million documents. In addition to the optional hard negatives, we also leverage in-batch, task-homogeneous negatives. In line with prior research (Li et al., 2023a), we employ task-specific instructions and temperature-scaled sampling of the datasets. Complete details about the data, training, and evaluation can be found in Appendix D.

**Results**  We found that training for more than 2,000 steps resulted in minimal performance gains. Table 4 presents the complete MTEB-English evaluation of all fine-tuned models. Although NeoBERT is $100M$ parameters smaller than all *large* baselines, it is the best model overall with a $+4.5\%$ relative increase over the second best model, demonstrating the benefits of its architecture, data, and pre-training improvements.

Table 4: MTEB(eng, v1) scores after 2,000 steps of fine-tuning with our unified contrastive learning.

| Size | Model | Class. 12 tasks | Clust. 11 tasks | PairClass. 3 tasks | Rerank. 4 tasks | Retriev. 15 tasks | STS 10 tasks | Summ. 1 tasks | Avg. |
|---|---|---|---|---|---|---|---|---|---|
| Base | BERT | 60.6 | 37.0 | 71.5 | 48.9 | 28.3 | 69.9 | 31.1 | 48.1 |
| | RoBERTa | 58.7 | 36.7 | 71.2 | 49.8 | 26.9 | 71.8 | 29.1 | 47.7 |
| | DeBERTaV3 | 45.9 | 15.2 | 44.3 | 39.0 | 3.5 | 42.2 | 25.0 | 26.9 |
| | NomicBERT$_{2048}$ | 55.0 | 35.3 | 69.0 | 48.8 | 30.5 | 70.1 | 30.1 | 47.1 |
| | ModernBERT | 58.9 | 38.1 | 63.8 | 48.5 | 21.0 | 66.2 | 30.1 | 45.0 |
| Medium | NeoBERT$_{4096}$ | 61.6 | **40.8** | **76.2** | 51.2 | **31.6** | **74.8** | 30.7 | **51.3** |
| Large | BERT | 59.8 | 39.3 | 70.9 | 49.7 | 29.6 | 71.4 | **31.2** | 49.1 |
| | RoBERTa | 57.1 | 39.3 | 72.5 | **51.3** | 30.0 | 71.7 | 31.1 | 48.9 |
| | DeBERTaV3 | 45.9 | 16.4 | 45.0 | 40.8 | 4.0 | 40.1 | 29.9 | 27.1 |
| | ModernBERT | **62.4** | 38.7 | 65.5 | 50.1 | 23.1 | 68.3 | 27.8 | 46.9 |

### 5.2.2 Contextual Document Embeddings

While a unified fine-tuning framework enables fair model comparisons, computationally intensive fine-tuning techniques are necessary to achieve top-ranking MTEB scores. We apply one such strategy, Contextual Document Embeddings (CDE) (Morris & Rush, 2024), to NeoBERT. CDE previously led the MTEB leaderboard with NomicBERT (cde-v1) and ModernBERT (cde-v2) for English models under 400M parameters. NeoBERT outperforms both, achieving an average score of **66.60**, the highest in its parameter class. See Appendix G for more details on the fine-tuning.

Table 5: MTEB(eng, v1) scores after fine-tuning with CDE. *Ranks on MTEB(eng, v1) for models under 400M parameters as of April 2025.

| Model | Class. 12 tasks | Clust. 11 tasks | PairClass. 3 tasks | Rerank. 4 tasks | Retriev. 15 tasks | STS 10 tasks | Summ. 1 tasks | Avg. | Rank* |
|---|---|---|---|---|---|---|---|---|---|
| NomicBERT | 81.72 | 48.32 | 84.69 | 56.75 | 53.27 | 81.64 | 31.23 | 65.00 | 3 |
| ModernBERT | 80.62 | 49.48 | 85.23 | 56.94 | 54.19 | **83.30** | **31.60** | 65.68 | 2 |
| NeoBERT | **82.14** | **50.29** | **86.71** | **57.71** | **56.37** | 82.30 | 30.98 | **66.60** | **1** |

### 5.3 Sequence Length

Following previous literature, NeoBERT underwent an additional 50k pre-training steps, during which it was exposed to extended sequences of up to 4,096 tokens. To assess the impact of this additional training, we randomly sampled 2,467 long sequences from the English subset of Wikipedia. For each sequence, we independently masked each token at position $i$ and computed its cross-entropy loss, $l_i$. The pseudo-perplexity of the entire sentence is then defined as $\mathcal{P} = \exp\left(\frac{1}{n}\sum_{i=1}^{n} l_i\right)$. We present the results in Figure 2.

Figure 2: Pseudo-Perplexity in function of the sequence length for NeoBERT$_{1024}$ *(left)* and NeoBERT$_{4096}$ *(right)*. This validates the effectiveness of the final pre-training stage on NeoBERT's ability to model long sequences.

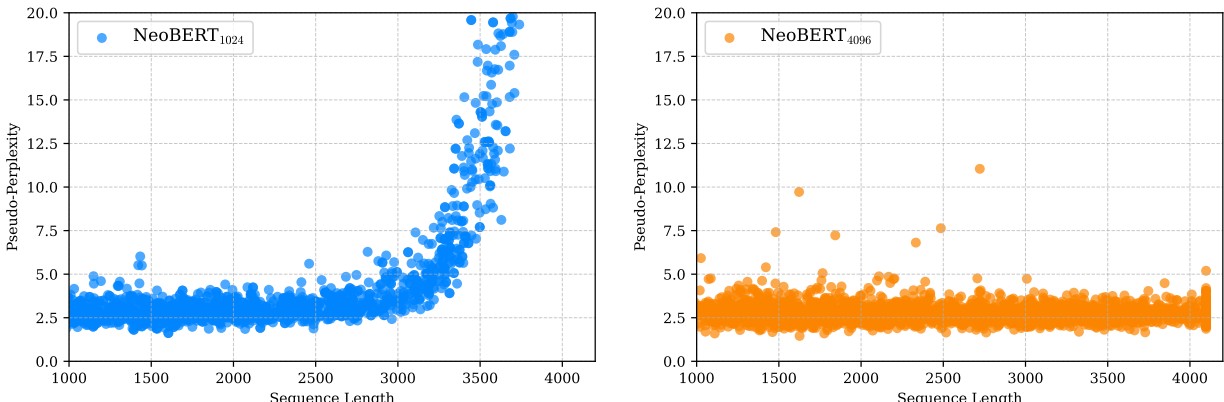

Although NeoBERT$_{1024}$ was trained exclusively on sequences of up to $1{,}024$ tokens, it generalizes effectively to context lengths approaching 3,000 tokens. This demonstrates the robustness of RoPE embeddings to out-of-distribution inputs. Moreover, after an additional 50k training steps with sequences up to $4{,}096$ tokens, NeoBERT$_{4096}$ successfully models longer sequences. This approach provides a compute-efficient strategy for extending the model's maximum context window beyond its original length. To further test NeoBERT's generalization capabilities, we extend the context beyond what was seen during training and present results in Appendix E.

### 5.4 Efficiency

To assess model efficiency, we construct a synthetic dataset consisting of maximum-length sequences of sizes $\{512, 1024, 2048, 4096, 8192\}$. For each sequence length, we scale the batch size from 1 to 512 samples or until encountering out-of-memory errors. Inference is performed for 100 steps on a single A100 GPU, and we report the highest throughput achieved for each model and sequence length. Figure 3 presents the results.

Due to their low parameter count and relatively simple architecture, BERT and RoBERTa are the most efficient for sequences up to 512 tokens. However, their use of positional embeddings prevents them from further scaling the context window. For extended sequences, NeoBERT significantly outperforms ModernBERT$_{base}$, despite having $100M$ more parameters, achieving a $46.7\%$ speedup on sequences of $4{,}096$ tokens.

## 6 Discussion

Encoders are compact yet powerful tools for language understanding and representation tasks. They require fewer parameters and significantly lower training costs compared to their decoder counterparts, making them strong alternatives for applications that do not involve text generation. Traditionally, the representational capacity of these models has been assessed through downstream tasks such as classification, in particular through the GLUE benchmark.

Figure 3: Model throughput (tokens per second) as a function of sequence length ($\uparrow$ is better). Above $1,024$ in sequence length, NeoBERT surpasses ModernBERT$_{base}$ despite having $100M$ more parameters.

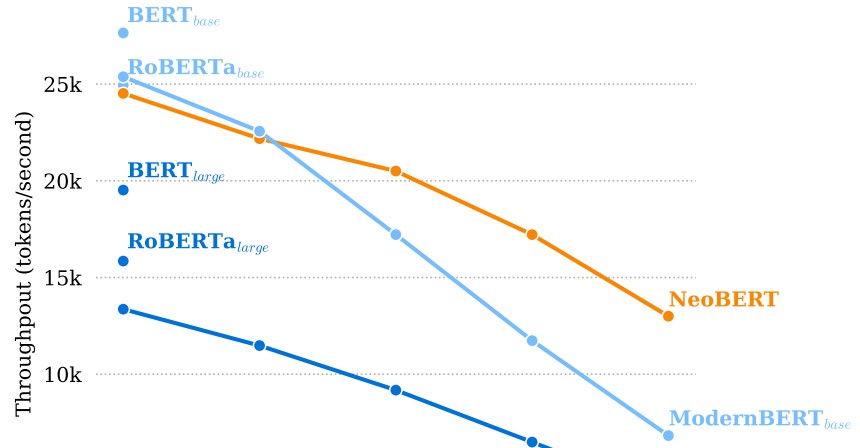

While GLUE has played a pivotal role in guiding model adoption, it includes only nine sequence classification datasets, four of which are entailment tasks. Moreover, its small dataset sizes and occasionally ambiguous labeling make it prone to overfitting, with models long surpassing human performance on the benchmark. Although DeBERTa-v3 achieves state-of-the-art performance on GLUE by a significant margin, our fine-tuning experiments reveal its comparatively poor performance on the more recent MTEB benchmark. MTEB encompasses a broader range of datasets and tasks, but attaining high performance on its leaderboard necessitates carefully crafted fine-tuning strategies with costly training requirements. As more complex fine-tuning strategies emerge, it becomes unclear what the source of score improvements is. Moreover, these strategies are not easily reproducible or accessible, limiting the possibility of fair comparison between pre-trained backbones.

This underscores the limitations of current evaluation paradigms and highlights the need for more affordable and standardized frameworks. We advocate for future research to focus on the development of standardized fine-tuning protocols and the exploration of new zero-shot evaluation methodologies to ensure a more comprehensive and unbiased assessment of encoder-only models.

## 7   Conclusion

We introduced NeoBERT, a state-of-the-art encoder pre-trained from scratch with the latest advancements in language modeling, architecture, and data selection. To ensure rigorous validation, we systematically evaluated every design choice by fully training and benchmarking ten distinct models in controlled settings. On GLUE, NeoBERT outperforms BERT$_{large}$ and NomicBERT and is comparable with RoBERTa$_{large}$ despite being 100M parameters smaller and supporting sequences eight times longer. To further validate its effectiveness, we conducted a comprehensive evaluation on MTEB, carefully isolating the effects of pre-training and fine-tuning to provide a fair comparison against the best open-source embedding models. Under identical fine-tuning conditions, NeoBERT consistently outperforms all baselines. In addition, when fine-tuned with more advanced techniques, NeoBERT ranks at the top of the MTEB leaderboard for models under 400M parameters. With its unparalleled efficiency, optimal depth-to-width, and plug-and-play compatibility, NeoBERT represents the next generation of encoder models. To foster transparency and collaboration, we release all code, data, model checkpoints, and training scripts, making NeoBERT the only fully open-source model of its kind.

**Broader Impact Statement**

Despite its improvements, NeoBERT inherits the biases and limitations of its pre-training data. Moreover, the greatest jump in performance stems from the pre-training dataset, and as newer, larger, and more diverse datasets become available, retraining will likely be needed to further improve its performance. Nonetheless, NeoBERT stands today as an affordable state-of-the-art pre-trained encoder with great potential for downstream applications.

## Acknowledgements

Sarath Chandar is supported by the Canada CIFAR AI Chairs program, the Canada Research Chair in Lifelong Machine Learning, and the NSERC Discovery Grant. Quentin Fournier is supported by the Lambda research grant program. The authors acknowledge the computational resources provided by Mila and the Royal Military College of Canada.

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

## A   Training details

NeoBERT was trained on 8 H100 for 1,050,000 steps, for a total of 6,000 GPU hours. In the first stage of training, we used a local batch size of 32, 8 gradient accumulation steps, and a maximum sequence length of $1,024$, for a total batch size of 2M tokens. In the second stage of training, we keep the theoretical batch size constant and increase the maximum sequence length to $4,096$.

## B   Ablations

### B.1   Baseline

Our first model, $M0$ is modeled after $\text{BERT}_{base}$ in terms of architecture. The only two differences are the absence of the next-sentence-prediction objective, as well as Pre-Layer Normalization. Each successive model, up until $M8$ is identical to the previous one on every point except for the change reported in Table 2.

### B.2   Tokenizers

We refer to the Llama Wordpiece tokenizer as llama-tok and the BERT BPE tokenizer as bert-tok. Under identical conditions between ablations $M2$ and $M3$, the use of llama-tok over bert-tok decreases the GLUE score by a relative drop of 2.9%. Although further investigation would be required to analyze this drop, we highlight several key differences between the two tokenizers.

While llama-tok uses Byte-Pair Encoding (BPE), bert-tok relies on the more expressive WordPiece algorithm, which better models token likelihoods in text. Moreover, their training data largely differs. llama-tok was trained on a dataset mostly composed of the CommonCrawl, whereas bert-tok used the higher-quality BookCorpus and WikiText datasets. Finally, llama-tok is cased and handles unknown UTF-8 characters at the byte level, while bert-tok is uncased and character-level. This leads llama-tok to include redundant tokens (e.g., "The" and "the") and rare byte sequences, limiting space for more frequent tokens under a finite capacity.

## C   GLUE

We perform a classical parameter search with learning rates in $\{5e-6, 6e-6, 1e-5, 2e-5, 3e-5\}$, batch sizes in $\{4, 8, 16, 32\}$ and weight decay in $\{1e-2, 1e-5\}$. In addition, we start training from the best MNLI checkpoint for RTE, STS, MRPC, and QNLI.

We fine-tune on the training splits of every glue dataset for 10 epochs, with evaluation on the validation splits every $n$ steps, $n$ being defined as $\min(500, \text{len}(\text{dataloader}) \; // \; 10)$ with early stopping after 15 evaluation cycles if scores have not improved. Following BERT, we exclude WNLI from our evaluation[3]. For tasks with two scores and for MNLI matched and mismatched, we report the average between the two metrics.

## D   MTEB

### D.1   Evaluation of pre-trained models

As demonstrated in Figure 4, evaluating out-of-the-box pre-trained models on MTEB is inconclusive. In that setting, $\text{BERT}_{base}$ outperforms both $\text{BERT}_{large}$ and $\text{RoBERTa}_{large}$, highlighting the importance of fine-tuning to ensure representative evaluation on the MTEB benchmark.

### D.2   Contrastive learning

Following the existing literature, we designed a simple fine-tuning strategy entirely agnostic to the models evaluated. We used cosine similarity and $\tau = 0.07$ as a temperature parameter in the contrastive learning

---

[3]See 12 in https://gluebenchmark.com/faq

| Model | Task | Batch Size | Learning Rate | Weight Decay |
|---|---|---|---|---|
| NeoBERT$_{1024}$ | CoLA | 4 | 6e-6 | 1e-5 |
| | MNLI | 16 | 6e-6 | 1e-2 |
| | MRPC | 8 | 2e-5 | 1e-5 |
| | QNLI | 8 | 5e-6 | 1e-5 |
| | QQP | 32 | 1e-5 | 1e-2 |
| | RTE | 8 | 6e-6 | 1e-5 |
| | SST-2 | 16 | 1e-5 | 1e-5 |
| | STS-B | 8 | 1e-5 | 1e-2 |
| NeoBERT$_{4096}$ | CoLA | 8 | 8e-6 | 1e-5 |
| | MNLI | 16 | 5e-6 | 1e-5 |
| | MRPC | 2 | 1e-5 | 1e-5 |
| | QNLI | 8 | 5e-6 | 1e-5 |
| | QQP | 32 | 8e-6 | 1e-5 |
| | RTE | 32 | 5e-6 | 1e-5 |
| | SST-2 | 32 | 8e-6 | 1e-2 |
| | STS-B | 32 | 2e-5 | 1e-5 |

Table 6: Optimal hyperparameters for GLUE tasks. The grid search was conducted over batch sizes $\{2, 4, 8, 16, 32\}$, learning rates $\{5e-6, 6e-6, 8e-6, 1e-5, 2e-5, 3e-5\}$, and weight decay values $\{1e-2, 1e-5\}$.

Figure 4: Zero-shot evaluation of BERT and RoBERTa on the English subset of MTEB.

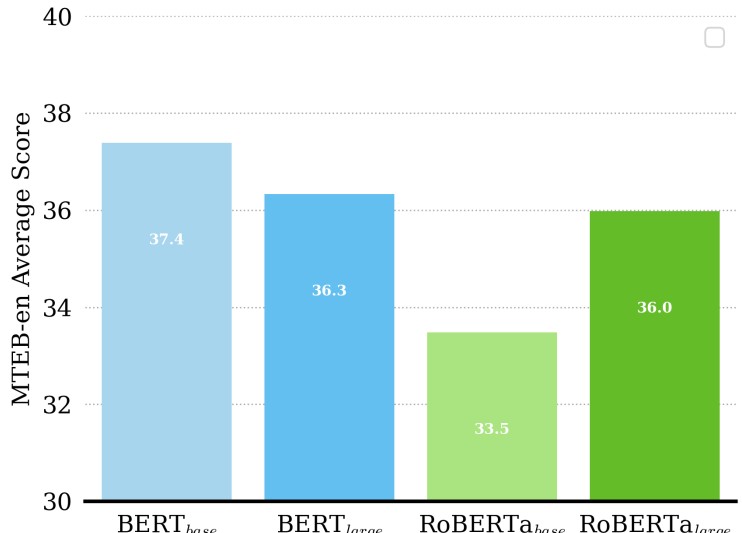

loss. Additionally, we sampled datasets with a multinomial distribution based on their sizes $(n_j)_{j=1}^m$ with $\alpha = 0.5$:

$$\pi = \frac{n_i^\alpha}{\sum_{j=1}^m n_j^\alpha}$$

We trained on the following fully-open datasets: AG-News (Zhang et al., 2016), All-NLI (Bowman et al., 2015; Williams et al., 2018), AmazonQA (Gupta et al., 2019), ConcurrentQA (Arora et al., 2022), GitHub

Issues (Li & Li, 2023), GooAQ (Khashabi et al., 2021), MedMCQA (Pal et al., 2022), NPR[4], PudMedQA (Jin et al., 2019), SentenceCompression (Filippova & Altun, 2013) StackExchange[5], TriviaQA (Han et al., 2019), Wikihow (Koupaee & Wang, 2018), Yahoo! Answers (Zhang et al., 2016) as well as the available training splits of MTEB datasets (StackOverFlowDupQuestion, Fever (Thorne et al., 2018), MS MARCO (Bajaj et al., 2018), STS12, and STSBenchmark (Cer et al., 2017)).

We fine-tune every model for 2,000 steps and evaluate on MTEB in float16. The complete results are presented in Table 4.

Figure 5: Average MTEB scores of fine-tuned encoders grouped by task type. The average score is computed across the 56 tasks of MTEB-English. NeoBERT is the best model on five out of seven task types and the best model overall. See Table 4 for complete scores.

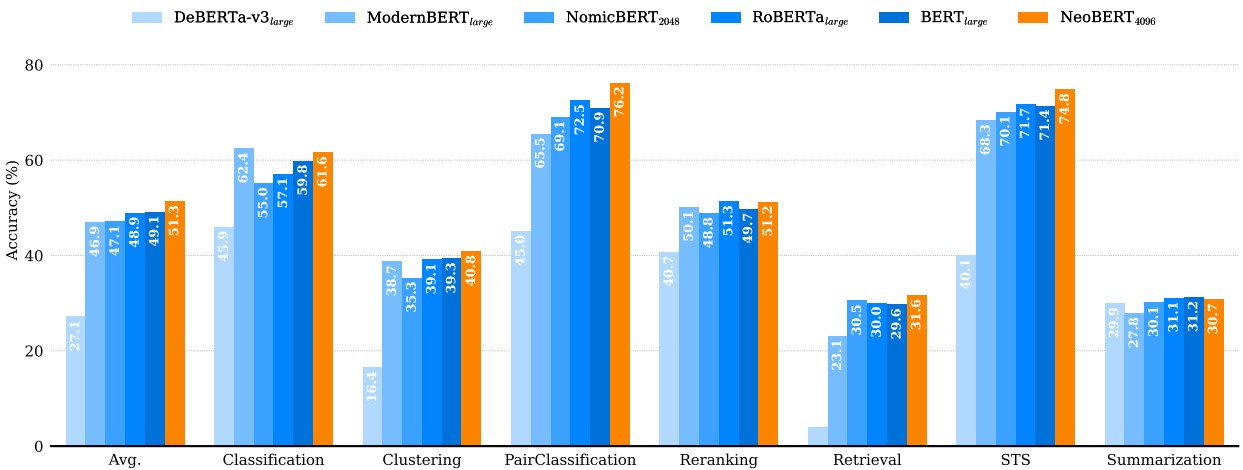

## D.3 Task instructions

We provide the set of instructions used for fine-tuning in Table 7 and evaluation in Table 8 and Table 9.

---

[4]https://huggingface.co/datasets/sentence-transformers/npr
[5]https://huggingface.co/datasets/sentence-transformers/stackexchange-duplicates

Table 7: Instructions for fine-tuning on the different contrastive learning datasets.

| Dataset | Instruction |
|---|---|
| AGNEWS | Given a news title, retrieve relevant articles. |
| ALLNLI | Given a premise, retrieve a hypothesis that is entailed by the premise. |
| AMAZONQA | Given a question, retrieve Amazon posts that answer the question. |
| CONCURRENTQA | Given a multi-hop question, retrieve documents that can help answer the question. |
| FEVER | Given a claim, retrieve documents that support or refute the claim. |
| GITHUBISSUE | Given a question, retrieve questions from Github that are duplicates to the given question. |
| GOOAQ | Given a question, retrieve relevant documents that best answer the question. |
| MEDMCQA | Given a medical question, retrieve relevant passages that answer the question. |
| MEDMCQA$_{CLUST}$ | Identify the main category of medical exams based on their questions and answers. |
| MSMARCO | Given a web search query, retrieve relevant passages that answer the query. |
| NPR | Given a news title, retrieve relevant articles. |
| PAQ | Given a question, retrieve Wikipedia passages that answer the question. |
| PUBMEDQA | Given a medical question, retrieve documents that best answer the question. |
| QQP | Given a question, retrieve questions from Quora forum that are semantically equivalent to the given question. |
| SENTENCECOMP | Given a sentence, retrieve semantically equivalent summaries. |
| STACKEXCHANGE | Given a Stack Exchange post, retrieve posts that are duplicates to the given post. |
| STACKOVERFLOW | Retrieve duplicate questions from StackOverflow forum. |
| STS12 | Retrieve semantically similar text. |
| STSBENCHMARK | Retrieve semantically similar text. |
| TRIVIAQA | Given a question, retrieve documents that answer the question. |
| WIKIHOW | Given a Wikihow post, retrieve titles that best summarize the post. |
| YAHOO$_{CLUST}$ | Identify the main topic of Yahoo posts based on their titles and answers. |

Table 8: Instructions for evaluation on the different MTEB tasks.

| Task name | Instruction |
|---|---|
| DBPedia | Given a query, retrieve relevant entity descriptions from DBPedia. |
| FEVER | Given a claim, retrieve documents that support or refute the claim. |
| FiQA2018 | Given a financial question, retrieve user replies that best answer the question. |
| HotpotQA | Given a multi-hop question, retrieve documents that can help answer the question. |
| MSMARCO | Given a web search query, retrieve relevant passages that answer the query. |
| NFCorpus | Given a question, retrieve relevant documents that best answer the question. |
| NQ | Given a question, retrieve Wikipedia passages that answer the question. |
| QuoraRetrieval | Given a question, retrieve questions that are semantically equivalent to the given question. |
| SCIDOCS | Given a scientific paper title, retrieve paper abstracts that are cited by the given paper. |
| SciFact | Given a scientific claim, retrieve documents that support or refute the claim . |
| Touche2020 | Given a question, retrieve detailed and persuasive arguments that answer the question. |
| TRECCOVID | Given a query on COVID-19, retrieve documents that answer the query. |
| SICK-R | Retrieve semantically similar text. |
| STS | Retrieve semantically similar text. |
| BIOSSES | Retrieve semantically similar text from the biomedical field. |
| SummEval | Given a news summary, retrieve other semantically similar summaries. |

Table 9: Instructions for evaluation on the different MTEB tasks.

| Task name | Instruction |
|---|---|
| AmazonCounterfactualClass. | Given an Amazon customer review, classify it as either counterfactual or not-counterfactual. |
| AmazonPolarityClass. | Given an Amazon review, classify its main sentiment into positive or negative. |
| AmazonReviewsClass. | Given an Amazon review, classify it into its appropriate rating category. |
| Banking77Class. | Given a online banking query, find the corresponding intents. |
| EmotionClass. | Given a Twitter message, classify the emotion expressed into one of the six emotions: anger, fear, joy, love, sadness, and surprise. |
| ImdbClass. | Given an IMDB movie review, classify its sentiment into positive or negative. |
| MassiveIntentClass. | Given a user utterance, find the user intents. |
| MassiveScenarioClass. | Given a user utterance, find the user scenarios. |
| MTOPDomainClass. | Given a user utterance, classify the domain in task-oriented conversation. |
| MTOPIntentClass. | Given a user utterance, classify the intent in task-oriented conversation. |
| ToxicConversationsClass. | Given comments, classify them as either toxic or not toxic. |
| TweetSentimentExtractionClass. | Given a tweet, classify its sentiment as either positive, negative, or neutral. |
| <dataset>ClusteringP2P | Identify the main and secondary category of <dataset> papers based on their titles and abstracts. |
| <dataset>ClusteringS2S | Identify the main and secondary category of <dataset> papers based on their titles. |
| <dataset>Clustering | Identify the topic or theme of <dataset> posts based on their titles. |
| TwentyNewsgroupsClustering | Identify the topic or theme of the given news articles. |
| SprintDuplicateQuestions | Retrieve duplicate questions from Sprint forum. |
| TwitterSemEval2015 | Given a tweet, retrieve tweets that are semantically similar. |
| TwitterURLCorpus | Given a tweet, retrieve tweets that are semantically similar. |
| AskUbuntuDupQuestions | Retrieve duplicate questions from AskUbuntu forum. |
| MindSmallReranking | Given a user browsing history, retrieve relevant news articles. |
| SciDocsRR | Given a title of a scientific paper, retrieve the relevant papers. |
| StackOverflowDupQuestions | Retrieve duplicate questions from StackOverflow forum. |
| ArguAna | Given a claim, find documents that refute the claim. Document |
| ClimateFEVER | Given a claim about climate change, retrieve documents that support or refute the claim. |
| CQADupstackRetrieval | Given a question, retrieve detailed question descriptions from Stackexchange that are duplicates to the given question. |

# E   Context Length

We compute the Pseudo-Perplexity on sequences sampled from English Wikipedia. Although NeoBERT has only been exposed to sequences up to 4,096 tokens during training, it correctly models sequences up to 6,000 tokens. Additional training or RoPE extension techniques, such as YaRN (Peng et al., 2023), would be needed to further extend the context window.

Figure 6: Pseudo-Perplexity in function of the sequence length for NeoBERT$_{4096}$ for extension beyond sequence lengths seen in training. NeoBERT generalizes natively for sequences under 6,000 tokens.

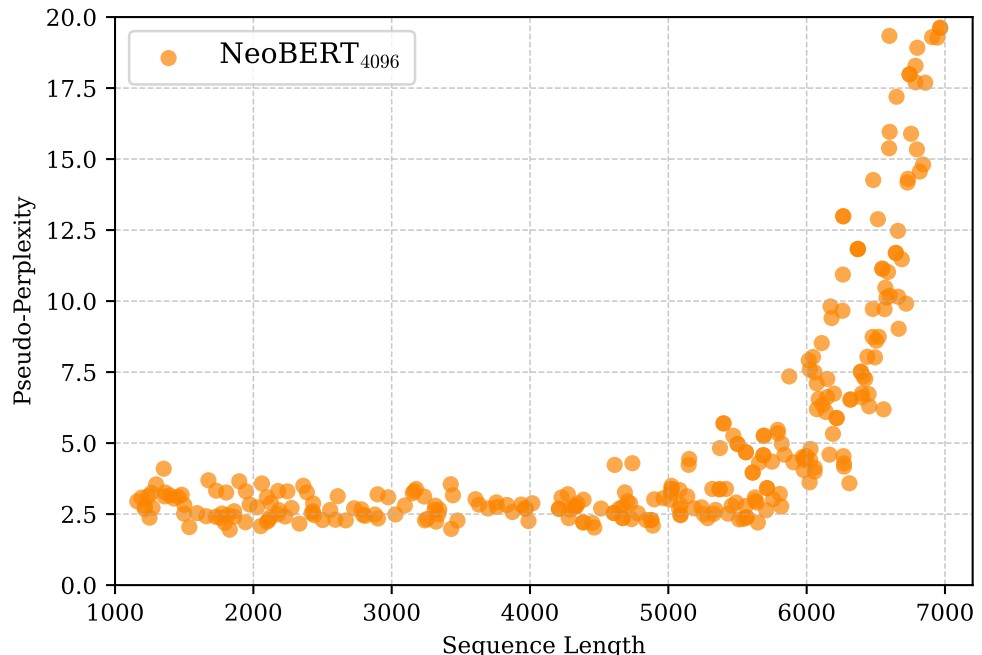

# F   Efficiency

Table 10 presents the complete results of model efficiency evaluations.

Table 10: Throughput ($10^3$ tokens / second) in function of the sequence length, with optimal batch size.

| Size | Model | 512 | 1024 | 2048 | 4096 | 8192 |
|---|---|---:|---:|---:|---:|---:|
| Base | BERT$_{base}$ | **27.6** $\pm$ 3.6 | - | - | - | - |
| | RoBERTa$_{base}$ | 24.9 $\pm$ 3.0 | - | - | - | - |
| | ModernBERT$_{base}$ | 25.4 $\pm$ 2.3 | **22.6** $\pm$ 2.7 | 17.2 $\pm$ 1.7 | 11.7 $\pm$ 0.8 | 6.8 $\pm$ 0.2 |
| Medium | NeoBERT | 24.5 $\pm$ 1.4 | **22.2** $\pm$ 1.7 | **20.5** $\pm$ 1.6 | **17.2** $\pm$ 1.2 | **13.0** $\pm$ 0.2 |
| Large | BERT$_{large}$ | 19.5 $\pm$ 0.6 | - | - | - | - |
| | RoBERTa$_{large}$ | 15.9 $\pm$ 0.3 | - | - | - | - |
| | ModernBERT$_{large}$ | 13.4 $\pm$ 0.2 | 11.4 $\pm$ 1.1 | 9.2 $\pm$ 0.7 | 6.5 $\pm$ 0.3 | 3.8 $\pm$ 0.1 |

# G   Contextual Document Embeddings

One popular application for BERT-style models is to generate embedding vectors for use in tasks such as classification, clustering, and retrieval. State-of-the-art embedding models below 400M parameters are

trained using contextual embedding techniques (Morris & Rush, 2024). The highest-performing similar-sized embedding models were trained with backbones initialized with NomicBERT (Nussbaum et al., 2024) and more recently ModernBERT (Warner et al., 2024). In this section, we question whether contextual document embeddings (CDE) might perform better when initialized from NeoBERT.

## G.1 Method

We train our embedding models using both the contextual batching strategy and contextual architecture described in Morris & Rush (2024).

**Contextual batching.** To train a contextual embedding model, we have to first group documents from a larger corpus into miniature 'pseudo-contexts' which comprise the batches for training. We do so by minimizing the following batch-clustering objective:

$$\min_{\substack{(\mathcal{B}^1,...\mathcal{B}^B) \\ (c^1,...,c^B)}} \sum_b \sum_{(d,q)\in\mathcal{B}^b} M((d,q),c^b)$$

Here, $\mathcal{B}_1...\mathcal{B}^B$ represent the orderings of batches of training data and $c^*$ are the corresponding centroids; $M(d,q)$ indicates the distance (typically Euclidean) between embeddings of document $d$ and query $q$ from batch $\mathcal{B}^b$.

We can approximate this efficiently using a modified K-Means solver. We follow the same procedure as Morris & Rush (2024) to cluster documents for each training set.

**Contextual architecture.** Contextual embedding models in effect use *two* model backbones: one to embed similar documents in context and another to embed documents conditioned on contextual information. We can express contextual embedding vector $\phi$ using both models $M_1$ and $M_2$:

$$\phi(x;\mathcal{D}) = M_2(M_1(d^1),\ldots,M_1(d^J),E(x_1),\ldots,E(x_T))$$

where $x$ is a document or query comprised of tokens $x_1...x_T$ and $\mathcal{D}$ is a set of documents from the surrounding context. We train $\phi$ end-to-end by backpropagating through both $M_1$ and $M_2$ using the custom two-stage gradient-caching procedure described in Morris & Rush (2024).

## G.2 Data and hyperparameters

**Training stages and data.** We train our models in two stages. The first stage is a large, noisy "contrastive pre-training" stage with $235M$ query-to-document pairs per epoch. The second is a short fine-tuning step on 1.5M high-quality datapoints. We train each stage for three epochs with the Adam optimizer with 1000 steps of warmup to a learning rate of $5e-5$ and linear decay to 0 throughout training.

Instead of mining hard negatives, we cluster the datasets for contextual batching. We embed for clustering and filter hard negatives within a batch using GTE-large (Li et al., 2023a). We set both the batch size and cluster size to 512. Unlike many contrastive training setups, we neglect to share negatives between GPUs.

We evaluate our contextual embedding models on the 56 tasks of the English version of MTEB (Muennighoff et al., 2023). To maintain consistency with prior iterations of CDE, we train these models with 512 in-context documents and a per-document sequence length of 512; this produces a second-stage input size of 1024 tokens.

## G.3 Results

Our contextual embedding results are detailed in Table 5, averaged across tasks in MTEB. With a score of 66.60, NeoBERT significantly outperforms ModernBERT in embedding-related tasks; in our case, the gap between NeoBERT and ModernBERT (about 1 MTEB point) is notably larger than the gap between ModernBERT and NomicBERT (approximately 0.6).

NeoBERT's largest improvements are shown in retrieval, reranking, clustering, and classification tasks. We see a small drop in performance in STS compared to ModernBERT, which we hypothesize may relate to the mix of pretraining data.

When compared to other models in MTEB, NeoBERT fine-tuned with CDE is state-of-the-art in its parameter class, outperforming all other similar-sized embedding models as of April 2025, such as GTE with 64.36 (Li et al., 2023a), BGE with 63.99 (Chen et al., 2024), and GIST with 64.13 (Solatorio, 2024).

