# OpenReview forum: "NeoBERT: A Next Generation BERT"
_TMLR — Accepted by TMLR_

### Review · Reviewer_N8Qd · 2025-03-07

**Summary Of Contributions:**

This paper introduces a new bidirectional Transformer-based encoder model, NeoBERT, and its full training recipe. The model is easy to plug-and-play and can replace other encoder models in deployment.
The training recipe of the model benefits from careful designs such as data curation, optimal depth-to-width ratio, and context length extension, thus achieving better performance and efficiency.
Its full training recipe documents extensive details for people to reproduce the model.
The authors also carried out ablation studies to assess the impact of each key component of the recipe.
All code, data, checkpoints, and training scripts are publicly released, facilitating adoption in both academic and industrial settings.

**Audience:**

Yes

**Broader Impact Concerns:**

No concerns on the ethical implications of the work.

**Claims And Evidence:**

Yes

**Requested Changes:**

What will significantly strengthen this paper is an in-depth discussion about why NeoBERT underperforms models of comparable sizes (e.g., bilingual-embedding-base, multilingual-e5-base) on MTEB Leaderboard.

**Strengths And Weaknesses:**

**Strengths**

Better encoder models are useful in both academia and industry settings. This paper delivers a strong but compact Transformer-based encoder model with long context window and high efficiency (high throughput), and thus is a solid contribution to the community.

The paper is clearly written.

Extensive experiments are carried out, and results are convincing.

The full training recipe is released, and thus it facilitates open-source research in this field.

**Weaknesses**

The main weakness of this work is the lack of discussion about why it underperforms certain models of comparable sizes on MTEB; see MTEB Leaderboard for those models: https://huggingface.co/spaces/mteb/leaderboard

Since the main contribution of this paper is a trained model with its recipe, but not a novel training method or so, I think that we need to see stronger results over competitors of comparable sizes, or an in-depth discussion about why it underperforms them, hopefully with a discussion about why this model is still better than those in certain ways.

---

> ### Author Response · Authors · 2025-04-23
> **Author response to Reviewer N8Qd**
>
> We sincerely appreciate the reviewers' detailed feedback and suggestions. We are pleased that they found our work to be a significant contribution to the research community (N8Qd, 3wjq), that our experiments are extensive with convincing results (N8Qd, r7sv, 3wjq), and that our paper is clearly written (N8Qd). We have carefully considered the reviewers' comments and concerns, and addressed them below.
>
> ---
>
> **Comment.** “The main weakness of this work is the lack of discussion about why it underperforms certain models of comparable sizes on MTEB; see MTEB Leaderboard for those models: https://huggingface.co/spaces/mteb/leaderboard”
>
> **Answer.** Developing encoders for downstream tasks like those in the MTEB benchmark involves two stages. The first is pre-training, which focuses on masked language modeling and is the main subject of our paper. The second is fine-tuning, which requires distinct techniques such as contrastive learning and is often surprisingly costly, taking weeks of additional training on 8 x H100 GPUs [1] [2].
>
> Since MTEB scores greatly depend on the fine-tuning strategy used, it is challenging to separate the effects of the two stages. To address this, one of our contributions is to conduct a controlled evaluation of the different design choices by applying an identical fine-tuning strategy to all models. We designed this strategy to be cost-effective while adhering to the existing literature. In this setting, NeoBERT outperforms all baselines, and in that sense, it’s the fine-tuning strategy that underperforms those used in the MTEB leaderboard rather than NeoBERT itself.
>
> Still, we recognize the importance of demonstrating a model’s best achievable performance, so we fine-tuned NeoBERT using the current state-of-the-art strategy. With CDE, **NeoBERT now achieves a score of 66.6 on MTEB-english-v1, ranking as the top model under 400M parameters**. It is 50M parameters smaller than the second-best model, cde-small-v2, which scores 65.58 and uses ModernBERT as a pre-trained backbone.
>
> **Comment.** “What will significantly strengthen this paper is an in-depth discussion about why NeoBERT underperforms models of comparable sizes (e.g., bilingual-embedding-base, multilingual-e5-base) on MTEB Leaderboard.”
>
> **Answer.** We believe that the state-of-the-art performance of NeoBERT on MTEB-english-v1 addresses the comment. Still, we will update the manuscript to make the above discussion clearer, as we believe such standardized fine-tuning strategies are valuable to the field, even if they do not reach SoTA performance.
>
> ---
>
> [1] Contextual Document Embeddings, Morris et al, 2024
>
> [2] Towards General Text Embeddings with Multi-stage Contrastive Learning, Li et al, 2023

---

### Review · Reviewer_r7sv · 2025-03-11

**Summary Of Contributions:**

This paper introduces NeoBERT, a next-generation encoder designed to enhance bidirectional models by incorporating state-of-the-art architectural advancements, modern data, and optimized pre-training. Despite having only 250M parameters, NeoBERT outperforms BERT, RoBERTa, NomicBERT, and ModernBERT on the MTEB benchmark while maintaining seamless compatibility as a plug-and-play replacement. To encourage further research and adoption, this paper provides a comprehensive evaluation framework, analyzes performance on GLUE, and releases all code, data, and training resources.

**Audience:**

Yes

**Claims And Evidence:**

Yes

**Requested Changes:**

as above

**Strengths And Weaknesses:**

Pro:

- It's good that this paper releases the dataset, unlike modernBERT, which does not include data release—a missed opportunity.
- The experiment is extensive and effectively isolates different factors for analysis.

Con:

- Why can't the support context length be as large as modernBERT?
- What is the benefit of two-stage training? Why not train directly on a 4K context length?
- How about using the Muon Operator instead of AdamW?
- Figure 1 is confusing—is it cumulative or separate between M0 and M10?
- In Figure 6, I’m curious how far the network can generalize. Can you show results for 10K or even 20K context length?
- Can you also include modernBERT in Figure 3?

One additional question is about the number of parameters. Nowadays, autoregressive models like LLaMA can be trained with around 8 billion parameters, while most BERT models are only about 500 million. Is this a common convention? Why didn’t the author consider training a larger model with more parameters and a bigger dataset? Would BERT’s performance continue to scale with an increasing number of parameters? I would appreciate it if the author could elaborate on this.

---

> ### Author Response · Authors · 2025-04-23
> **Author response to Reviewer r7sv**
>
> We sincerely appreciate the reviewers' detailed feedback and suggestions. We are pleased that they found our work to be a significant contribution to the research community (N8Qd, 3wjq), that our experiments are extensive with convincing results (N8Qd, r7sv, 3wjq), and that our paper is clearly written (N8Qd). We have carefully considered the reviewers' comments and concerns, and addressed them below.
>
> ---
>
> **Comment.** Why can't the support context length be as large as modernBERT?
>
> **Answer.** In Transformers, the attention computation scales quadratically in sequence length. As a result, training on sequences of length 8192 is 4 times more expensive than on length 4096. With our available compute, we had to arbitrate between the training scale and size of the context window. As most of the sentences in the training corpus and downstream tasks are under 1024 in length, we chose a context window of 4k as a trade-off between effective representation of long sequences and training for a large amount of tokens.
>
> **Comment.** What is the benefit of two-stage training? Why not train directly on a 4K context length?
>
> **Answer.** Training the model on length 4K from scratch would be 16 times more expensive than on length 1k. Two-stage training is a standard way to overcome this [1] [2]. The majority of computation and semantics learning happens on short context. In the second phase, 50k additional steps allow generalization to longer sequences (Fig 2). In the end, the model supports sequences of identical length while requiring significantly less computation than if it had been trained on a 4K context from scratch.
>
> **Comment.** How about using the Muon Operator instead of AdamW?
>
> **Answer.** We recognize that the Muon optimizer promises improved training speed. However, we would like to highlight that there have been questions regarding its efficacy when finetuned with AdamW [3]. Moreover, when we started the project, its implementation was not compatible with distributed training [4]. As of now, it is still not compatible with DeepSpeed. Since we trained in a distributed environment, our trade-off for speed was in favor of keeping AdamW rather than having to stop using DeepSpeed Zero2.
>
> **Comment.** Figure 1 is confusing—is it cumulative or separate between M0 and M10?
>
> **Answer.** We have modified the caption of Figure 1 to alleviate the ambiguity. Each of the bars represents an individual GLUE score for models that have cumulative modifications from M0 to M10.
>
> **Comment.** In Figure 6, I’m curious how far the network can generalize. Can you show results for 10K or even 20K context length?
>
> **Answer.** We have tested the generalization capabilities of NeoBERT and added these results to Appendix E. Thanks to RoPE, NeoBERT effectively generalizes up to 6,000 tokens. An additional extension phase or other techniques such as YaRN [5] would be necessary to extend beyond this.
>
> **Comment.** Can you also include modernBERT in Figure 3?
>
> **Answer.** ModernBERT is already included in Figure 3, the light blue line being ModernBERT base and the dark blue line ModernBERT large. We have improved the clarity of the figure and legend.
>
> **Comment.** One additional question is about the number of parameters. (...) Is this a common convention? Why didn’t the author consider training a larger model with more parameters and a bigger dataset? (...)
>
> **Answer.** Yes, encoders are typically much smaller than decoders, which is one of their key advantage, for two reasons.
> First, unlike encoders, decoders rely only on left context and provide generation capacities, thus requiring richer representations compared to an encoding task. This allows encoders to be much more compact, with limited benefits from scaling. For instance, Nvidia tried scaling BERT to 4B parameters [6], but only offered improvements of 1.2% on MNLI and 0.7% on QQP over models 10 times smaller.
> Second, while scaling laws give the optimal size and training time for a fixed compute budget, they do not account for the utilization cost. In our case, by training a larger model for a shorter amount of time, we could have reached identical scores with a lower compute budget. Instead, we decided to train a smaller model for longer to guarantee low inference costs for users.
>
> ---
>
> **Additional results**
>
> We would like to present new results produced in response to reviewer #N8Qd. We fine-tuned NeoBERT using CDE [7], a SoTA contrastive learning strategy. With CDE, **NeoBERT now achieves a score of 66.6 on MTEB-english-v1, ranking as the top model under 400M parameters**. It is 50M parameters smaller than the second-best model, cde-small-v2, which scores 65.58 and uses ModernBERT as a pre-trained backbone.
>
> ---
>
> [1] Grattafiori et al, 2024
>
> [2] Yang et al, 2024
>
> [3] Liu et al, 2025
>
> [4] https://github.com/KellerJordan/Muon/pull/3
>
> [5] Peng et al, 2023
>
> [6] Shoeybi et al, 2019
>
> [7] Morris et al, 2024

---

> > ### Comment · Reviewer_r7sv · 2025-04-23
> > **Thank You!**
> >
> > ty

---

### Review · Reviewer_3wjq · 2025-04-13

**Summary Of Contributions:**

This paper introduces NeoBERT, a next-generation encoder model designed to bridge the gap between modern autoregressive language models and traditional encoders like BERT and RoBERTa. The authors address the fact that while decoders have seen significant advances in recent years, encoder architectures have remained relatively static despite their critical importance for downstream NLP tasks.
The key contributions include:

- A comprehensive redesign of the encoder architecture incorporating modern techniques such as RoPE positional embeddings, optimal depth-to-width ratio, SwiGLU activation, and Pre-RMSNorm.
- Training on RefinedWeb, a modern large-scale dataset (2.8TB), with an extended context window of 4,096 tokens.
- Extensive ablation studies conducted by fully pre-training multiple models with different configurations, allowing for precise isolation of the impact of each modification.

**Audience:**

Yes

**Broader Impact Concerns:**

N/A.

**Claims And Evidence:**

Yes

**Requested Changes:**

- Provide more justification for using AdamW with cosine decay instead of WSD, which has become standard in modern language models including ModernBERT. Was this an empirical choice, or were there other considerations?
- Include more detailed experiments on RoPE's extrapolation capabilities, especially since this is one of RoPE's key advantages. While Figure 2 shows some analysis of sequence length modeling, specific extrapolation experiments would enhance the contribution.
- Provide more information about the training infrastructure and costs to help other researchers better understand the resources required to replicate or build upon this work.

**Strengths And Weaknesses:**

### Strengths

- The paper includes extensive ablations by fully training 10 different models, which is computationally expensive but provides valuable insights into the impact of each design choice.
- By releasing all code, data, and model checkpoints, the authors make a significant contribution to the research community and enhance reproducibility.
- The authors balance performance improvements with practical considerations like inference speed and computational efficiency, making the model more accessible for real-world applications.

### Weaknesses

- The authors use AdamW with cosine decay rather than the WSD scheduler used in ModernBERT and other recent LLMs. The rationale for this choice isn't fully explained. Since the authors will release their checkpoints, using WSD could potentially benefit users who want to fine-tune the model on their own data mixes by providing better annealing capabilities.
- While the authors note that using the LLaMA BPE tokenizer resulted in a 2.1% performance decrease, there isn't a deeper investigation into why this occurred.
- Given that RoPE theoretically allows for better extrapolation to longer sequences, more detailed experiments on zero-shot context length extrapolation would have strengthened the paper.

---

> ### Author Response · Authors · 2025-04-23
> **Author response to Reviewer 3wjq**
>
> We sincerely appreciate the reviewers' detailed feedback and suggestions. We are pleased that they found our work to be a significant contribution to the research community (N8Qd, 3wjq), that our experiments are extensive with convincing results (N8Qd, r7sv, 3wjq), and that our paper is clearly written (N8Qd). We have carefully considered the reviewers' comments and concerns, and addressed them below.
>
> ---
>
> **Comment.** Provide more justification for using AdamW with cosine decay instead of WSD, which has become standard in modern language models including ModernBERT. Was this an empirical choice, or were there other considerations?
>
> **Answer.** We acknowledge that the WSD scheduler provides interesting benefits of continuing training for end users. However, recent research provides evidence that theoretically optimal learning rates have a shape closely resembling that of the cosine decay schedule (see Section 6 of [1]). Moreover, although there has been growing interest in WSD, most modern LLMs still rely on linear warmup followed by cosine decay [2] [3] [4]. Having limited compute, we made a conservative choice to stick to that learning rate schedule. In the future, it would definitely be valuable to experiment with how WSD translates to encoders, and we would be interested to see an analysis on the topic.
>
> **Comment.** While the authors note that using the LLaMA BPE tokenizer resulted in a 2.1% performance decrease, there isn't a deeper investigation into why this occurred.
>
> **Answer.** We agree that it is valuable to understand the reasons for the performance gaps between the tokenizers. In Appendix B.2, we have added a discussion on key differences between the LlaMA (llama-tok) and BERT (bert-tok) tokenizers. First, while llama-tok was trained with BPE (byte-pair encoding), bert-tok was trained with the slightly more sophisticated WordPiece algorithm. Theoretically, WordPiece better matches the likelihood distribution of tokens in the corpus. Moreover, the training data of llama-tok is in large part the Common Crawl, which had lower quality filters as of the time of development (late 2022 - early 2023). In comparison, bert-tok was trained on a high-quality assembly of BookCorpus and WikiText. Finally, llama-tok is cased and byte-level for unknown utf-8 tokens, while bert-tok is uncased and character-level. Those characteristics imply that llama-tok has allocated space to tokens that are duplicates (“The” vs “the”) or non-frequent utf8 characters, effectively losing capacity for more frequent tokens. However, experimentally investigating the impact of each of these differences is outside the scope of our paper. [5] [6]
>
> **Comment.** Include more detailed experiments on RoPE's extrapolation capabilities, especially since this is one of RoPE's key advantages. While Figure 2 shows some analysis of sequence length modeling, specific extrapolation experiments would enhance the contribution.
>
> **Answer.** We have run additional extrapolation experiments and added these results to Appendix E. Thanks to RoPE, NeoBERT effectively generalizes to sentences over 5,000 tokens long.
>
> **Comment.** Provide more information about the training infrastructure and costs to help other researchers better understand the resources required to replicate or build upon this work.
>
> **Answer.**  We acknowledge the importance of fully open-sourcing training procedures as a contribution to the community. The training details can be found in Appendix A, and the complete training scripts and code will be released upon publication of the paper.
>
> —
>
> **Additional results**
>
> We would like to present new results produced in response to reviewer #N8Qd. We fine-tuned NeoBERT using CDE [7], a SoTA contrastive learning strategy. With CDE, **NeoBERT now achieves a score of 66.6 on MTEB-english-v1, ranking as the top model under 400M parameters**. It is 50M parameters smaller than the second-best model, cde-small-v2, which scores 65.58 and uses ModernBERT as a pre-trained backbone.
>
> —
>
> [1]  No Wrong Turns: The Simple Geometry Of Neural Networks Optimization Paths, Guille-Escuret, 2023
>
> [2] The Llama 3 Herd of Models, Grattafiori et al, 2024
>
> [3] 2 OLMo 2 Furious, Team OLMo et al, 2024
>
> [4] Qwen2 Technical Report, Yang et al, 2024
>
> [5] LLaMA: Open and Efficient Foundation Language Models, Touvron et al, 2023
>
> [6] BERT: Pre-training of Deep Bidirectional Transformers for Language Understanding, Devlin et al, 2018
>
> [7] Contextual Document Embeddings, Morris et al, 2024

---

### Decision · Action_Editor_5yUi · 2025-05-28

**Recommendation:** Accept as is

**Comment:**

This paper introduces NeoBERT, a next-generation encoder model designed to bridge the gap between modern autoregressive language models and traditional encoders like BERT and RoBERTa. After author rebuttal, it received 1 Accept and 2 Leaning Accept recommendations. Reviewers are generally happy about the paper, commenting that NeoBERT is a well-executed and timely contribution that modernizes BERT-style encoders with state-of-the-art design, extensive ablations, and strong open-source commitment. The authors address all concerns with additional experiments and achieve top-tier performance under fair evaluation. The author response clearly explains why their model underperforms models of comparable sizes, along with new results to show continued improvements.

Overall, given all the positive feedback and the timely contributions, the Action Editor would like to recommend acceptance of the paper.

**Audience:**

Yes.

**Claims And Evidence:**

Yes.